# Experimental Research on the Use of a Selected Multi-Criteria Method for the Cutting of Titanium Alloy with an Abrasive Water Jet

**DOI:** 10.3390/ma16155405

**Published:** 2023-08-01

**Authors:** Aleksandra Radomska-Zalas

**Affiliations:** Faculty of Technology, Jacob of Paradies University, 66-400 Gorzow Wielkopolski, Poland; aradomska-zalas@ajp.edu.pl

**Keywords:** abrasive water jet cutting process, advanced manufacturing process, VIKOR method

## Abstract

The use of selected multi-criteria decision methods for the optimization of cutting processes by abrasive water jet methods is increasingly being used in industrial processes. This is due to the complexity of the processes and the need to reduce operating costs. Process optimization methods are available to support organizational processes including the design phase, quality assurance, production automation, and many more. This article presents the current state of research on the water-abrasive cutting process and the use of multi-criteria methods in optimizing this process. This article presents a detailed methodological study of the VIKOR approach to optimization, indicating the applicability conditions, assumptions, and limitations on the example of high-pressure abrasive water jet cutting of elements made of titanium alloy utilizing HPX garnet abrasive. As a result of the research conducted, the best input parameters of the cutting process for abrasive flow rate, pressure, and the traverse speed of the cutting process were determined. The achieved result is consistent with the assumption that the most favorable output parameters are the highest cutting depth and the lowest level of roughness.

## 1. Introduction

Multi-criteria decision analysis (MCDA) methods have many applications in science and industry. In the literature, there is an abundance of research in the field of improving multi-criteria methods but there is little information on how to select a method for the decision-making process. One example is the research conducted by Dyckhoff and Souren, who reviewed the multi-criteria decision analysis (MCDA) and found no developed methodology that fully integrates MCDA into production processes [1]. Each method can give different recommendations and thus different results. For this reason, it is important to correctly define the evaluation criteria and their importance, i.e., the importance of a given process. Depending on the process that is being optimized, attention should be paid to elements such as cost, time, internal and external requirements, implementation potential, and others [2].

The use of multi-criteria methods is to determine the optimal parameters of the process, taking into account the output criteria. There are many process management methods in the business process of an enterprise, such as focusing on reliability, integrated product or service lifecycle, as described in detail by Grisold et al. [3]. These processes are tailored to the companies, their areas of operation, profitability, and human resources and require support to increase the efficiency of the organizations in which they are implemented [4]. However, since process support is still in the research phase, there are many gaps in modeling various aspects, such as the approach to industrial processes [5,6]. 

There are many studies on the application of multi-criteria methods in industrial processes, which have been written about by, among others, Kluczyk [6], Khalili and Duecker [7], and Petel et al. [8]. However, due to the complexity and diversity of industrial processes, there is still an important gap in the selection of an appropriate multi-criteria method for process optimization. 

Research was carried out and used in the process of cutting with a water-abrasive jet using such methods as: WASPAS [9], TOPSIS [10], and RSM [11]. They gave optimistic results, but their disadvantage is the use of a system of subjective weights [12]. Therefore, an attempt has been made to use the VIKOR method, which avoids the problem of multiple choice and results in the acceptance of conflicting norms by recognizing a consensus as sufficient [13]. Sayadi wrote that the result of the VIKOR method is based on the maximum group profit for the majority and the minimum loss for the minority [13]. The VIKOR index is designed to illustrate the closeness to an exemplary solution. In addition, it presents a real result that is close to the ideal, as well as illustrating the worst scenario that can occur [14]. 

The research process was automated by designing and implementing an IT system which, thanks to the implementation of multi-criteria methods, is designed to support the process of selecting the optimal solution [15]. The optimization IT system was created to increase the efficiency of enterprises by facilitating decision-making which, in the previous activities, was improperly organized at the level of company management. The purpose of creating and using an optimization decision support system is to collect individual data, i.e., data extraction from collections. It is also important that it provides access to data at any time of operation and allows for their easy interpretation. In order to increase the efficiency of operation, the system was equipped with multi-criteria decision support methods.

The task of the system is to identify a group of possible decisions and to present the consequences of a given choice by the use of computational and simulation models. The system output can choose a decision option based on the data input and the external conditions affecting the company [16]. The process of testing the system allows for the conclusion that, by using the designed system, the user has the ability to use the proposed formal procedures, and to choose one, best, solution to the problem under specific input conditions. The importance of the above factors was pointed out by Patyk [17]. 

High-pressure abrasive waterjet processing is one of the newest areas of advanced production processes. Ahmed et al. wrote that AWJ (abrasive water jet) cutting includes the erosion of the machining material by the abrasive grains in a high velocity water jet [18]. AWJ is generated by the passage of a pure water jet through a special mixing chamber to mix it with abrasive grains and achieve a high velocity (800 m/s) [19] and focus it in the water-abrasive nozzle, known as the focusing tube. The use of an abrasive water jet is a competitive solution to conventional material separation methods. This is mainly since it has a universal character related to the wide possibilities of cutting and drilling a range of materials, combined with the simultaneous lack of change in the structure of the materials subject to processing. 

The AWJ process is used to cut various materials. Xu et al. described the use of the AWJ method in wood–plastic composite cutting [20], where they indicated that the selection of appropriate parameters has an influence on obtaining a high-quality cutting surface. The basic machining parameters that have a decisive impact on shaping the geometry of the cutting gap include the water pressure, the traverse speed of the cutting head, the type and abrasive flow rate, the diameter and length of the focusing tube, stand-off distance, which they described, among others: Zhu et al., who described the process of cutting wood [21] and Aydin et al., who described the use of waste produced following the cutting of solid granite as an alternative abrasive in the water jet cutting of marble [22]. This process was also described by Kaczmarek, who wrote that the treatment of materials with a high-pressure water jet is a complex process [23]. To cut any material and to increase the efficiency of the process, dry natural abrasives such as garnet and olivine or synthetic abrasives (broken glass, aluminum oxide) are added to the jet [24]. The abrasive should be selected bearing in mind the criterion of the relationship between the life of the nozzle and the machining efficiency of the workpiece [25].

The AWJ process and its use of HPX have a long industrial history. However, the conducted research is intended to bring the wider possibilities of using HPX as an abrasive closer. It should be added that this article contains the results of the author’s system to support the optimization of the cutting process with an abrasive water jet using the VIKOR method. This method was used to test elements made of titanium in HPX abrasive technology. The analysis of the problem so far allows us to conclude that AWJ can be used for titanium machining. This is justified due to the cutting efficiency and the obtaining of better surface roughness properties of the cut groove. However, the use of multi-criteria methods, including VIKOR, for titanium cutting technology is not widely described in the literature. Therefore, the author decided to test the VIKOR method when cutting titanium using the HPX garnet abrasive. 

## 2. Materials and Method

### 2.1. Materials

The tests were carried out on samples of titanium alloy. Titanium and its alloys are characterized by high mechanical strength to mechanical loads, high hardness and flexibility, and, like most titanium alloys, exceptional resistance to corrosion in most natural environments [26] and many industrial environments [27]. The tests used titanium alloys known as Ti6Al4V, Ti6Al-4V or Ti 6-4. They have a two-phase structure and are the most popular of all the titanium alloys. Their chemical composition is presented in Table 1.

The density value of Grade 5 titanium is 4.42 kg/dm^3^ and its tensile strength value is 950 MPa. Additionally, the shear modulus is from 40 to 44 kN/mm^2^, the modulus of elasticity is from 105 to 120 kN/mm^2^ and the Vickers hardness value is 330 kG/mm^2^ [27].

It should be added, however, that while titanium has good properties in terms of practical applications, its processing is problematic. Since the material is characterized by high strength, with low thermal conductivity and chemical reactivity, especially under the influence of high temperatures, the service life of cutting blades is significantly reduced. In addition, the relatively low Young’s modulus for titanium alloys leads to material springback during machining, resulting in long, continuous chips that cause the cutting tool to curl [29]. Abrasive water jet (AWJ) technology provides effective machining of titanium alloys and eliminates these problems.

The Barton HPX80 garnet was selected as the abrasive material for the tests presented [30]. Garnet is a natural product, therefore its chemical composition may vary within certain limits. The average properties of this material are presented in Table 2. 

The HPX garnet from the American deposit in the Adirondack Mountains, New York State, is produced by crushing rock. The grains are characterized by very sharp, angular cutting edges which cut faster and offer better surface finishes [31]. The grains distribution of HPX80 garnet used in tests is presented in the Figure 1.

### 2.2. The VIKOR Method

The VIKOR method is a multi-criteria method included in the group of MCDM methods (multi-criteria decision making), which was introduced by Serafim Opricovic. Its task is to solve decision-making problems with conflicting and non-complementary criteria with the assumption that a compromise can be the solution [13]. In this method, the most optimal solution is closest to the ideal one, and the individual alternatives are evaluated for all the adopted criteria. The result of the VIKOR method is a ranked list of alternatives, and the compromise solution is the closest to the ideal [32]. 

The VIKOR method avoids the problem of multiple choice, which results in accepting conflicting norms by accepting the consensus as sufficient. The result is based on the maximum group gain for the majority and the minimum loss for the minority. The VIKOR index is intended to illustrate the proximity to an ideal solution. Moreover, this method shows a real result that is close to the ideal as well as the weakest one that can occur [14].

In the VIKOR approach, there are numerous alternatives from which it can be concluded whether the selection of given criteria can be considered optimal. To determine this, the obtained solutions should be compared and evaluated. This can be achieved by creating, in the first stage of the VIKOR model, the same matrix *D* as in determining the weights by the entropy method [11]. The relative decision matrix has dimension MxN and is expressed as follows (1):(1)$$D=\left[\begin{array}{ccc} x_{1,\;1} & x_{1,\;2} & x_{1,\;n}\\ x_{2,\;1} & x_{2,\;2} & x_{2,\;3}\\ x_{m,\;1} & x_{m,\;2} & x_{m,\;n} \end{array}\right]$$
where:*m*—number of options, *n*—number of initial conditions.

The decision matrix presented should be adjusted using the degrees reaching the maximum for the variables for which the largest possible values are sought, i.e., the so-called stimulants (2):(2)$$r_{i,j}=\frac{x_{i,j}-{\min x}_{i,j}}{{\max x}_{i,j}-{\min x}_{i,\;j}}$$
where:*r*_*i*,*j*_ is the normalized element of the decision matrix.

On the other hand, the variables that should take the smallest values which determine better process efficiency, i.e., the so-called destimulants, are calculated as follows (3):(3)$$r_{i,j}=\frac{{\max x}_{i,j}-x_{i,j}}{{\max x}_{i,j}-{\min x}_{i,\;j}}$$

As regards the individual criteria calculated on the basis of Equations (2) and (3) above, the outcome is as follows:(4)$$T_{i,j}=\frac{r_{i,j}}{\sum_{i=1}^{m}r_{i,j}}$$

If you want to determine the dispersion value referring to Equation (4), you should determine this value from the Equation (5):(5)$$e_{i,j}=-\frac{1}{\ln m}\;\sum_{i=1}^{n}T_{i,j}\ln T_{i,j}$$

Finally, having at our disposal the steps taken so far, we come to the point where it is possible to determine the weight for a given criterion, which is defined as follows:(6)$$w_{i}=\frac{1-e_{i,j}}{\sum_{i=1}^{n}\left(1-e_{i,j}\right)}$$

Having knowledge about determining the weight of entropy, one can proceed to the algorithm used in the VIKOR method. For this purpose, the normalized elements of the matrix should be determined, according to Equation (7):(7)$$f_{i,j}=\frac{x_{i,j}}{\sqrt{\sum_{i=1}^{m}x_{i,j}^{2}}}$$
where: *f*_*i*,*j*_—normalized matrix element,*x*_*i*,*j*_—element of the relative decision matrix.

The next step is to calculate the measure *S_i_*, using the equation for the favorable element, according to Equation (8):(8)$$S_{i}=\sum_{i=1}^{n}w_{i}\left[\frac{{\max f}_{i,j}-f_{i,j}}{{\max f}_{i,j}-{\min f}_{i,j}}\right]$$
where: *w_i_*—attribute weight.

Next is to calculate the measure *S_i_*, using the equation for the unfavorable element, according to Equation (9):(9)$$S_{i}=\sum_{i=1}^{n}w_{i}\left[\frac{f_{i,j}-min\;f_{i,j}}{{\max f}_{i,j}-{minf}_{i,j}}\right]$$

In the next step, the measure *R_i_* is calculated for the favorable variant (10) and the unfavorable variant (11):(10)$$R_{i}=\max\left\{w_{i}\;\left[\frac{{\max f}_{i,j}-f_{i,j}}{{\max f}_{i,j}-{\min f}_{i,j}}\right]\right\}$$
(11)$$R_{i}=\max\left\{w_{i}\;\left[\frac{f_{i,j}-{\min f}_{i,j}}{{\max f}_{i,j}-{\min f}_{i,j}}\right]\right\}$$

The last step in the VIKOR method is to determine the *Q_i_* coefficient according to the Equation (12):(12)$$Q_{i}=v\;\left[\frac{S_{i}-{\min S}_{i}}{{maxS}_{i}-{\min S}_{i}}\right]\;\left(1-v\right)\left[\frac{R_{i}-{\min R}_{i}}{{\max R}_{i}-{\min R}_{i}}\right]$$
where: *ν*—impact factor. The VIKOR coefficient can take any value between 0 and 1.

In the VIKOR method, the pattern consists of two solutions, i.e., the best alternative and the worst alternative [33]. Given a given decision possibility, a weighted average and a maximum weighted distance from the best result, as well as the result of the aggregate indicator, must be determined. Therefore, three structures are created to be successively juxtaposed by experimenting with a situation in which the advantage and stability of decisions are tolerated [34].

### 2.3. Experimental Setup and Equipment

The Abrasive Water Jet (AWJ) cutting process is based on the utilization of a high-pressure mixture water jet with the introduction of garnet or other abrasive grains which are mixed with the water jet in a special cutting head creating an abrasive water jet (AWJ). This jet can cut through all materials with high performance and quality. 

The AWJ cutting involves directing a jet at the workpiece and causing displacement relative to the material (Figure 2). 

To reach accurate cuts and shapes, the material or the AWJ nozzle can be moved in a CNC controlled manner. During the cutting process, the water takes away heat, reducing heat concentration and diminishing the potential for thermal deformations and failures of the material. Cutting tests were carried out on the OMAX 60120 JetMachining Center (OMAX, Kent, WA, USA) (Figure 3a), according to the L9 orthogonal array design model with the following control parameters: garnet mass flow rate, pressure, and traverse speed. The first output parameter was the maximum depth of cut, which unambiguously characterizes the performance of this material separation process. In industrial practice, a useful depth equal to, at most, half of the maximum depth is used (Figure 2). The second output parameter was the roughness of the cut surface, specifically the roughness factor Sq.

The rest of the values of the control parameters involved in the process were considered fixed. These included: -The diameter of the water nozzle: 0.3 mm,-The diameter of the water-abrasive nozzle: 0.76 mm,-The distance of the nozzle from material: 4 mm.

Roughness and cutting depth measurements and photographs of the tested samples were made using an OLYMPUS DSX 1000 digital microscope (OLYMPUS, Shinjuku, Japan) (Figure 3b) and the cut surface was observed and analysed on a Thermo Scientific Axia ChemiSEM Scanning Electron Microscope (Waltham, MA, USA) (Figure 3c). 

The DSX 1000 digital microscope enables a detailed analysis of the microstructure of various materials at a magnification of up to 7000x. The scanning electron microscope enables precise determination of the surface condition of the tested materials and identification of inclusions; additionally, the EDS attachment enables quantitative and qualitative analysis in relation to the elements present. Both microscopes are the property of the Faculty of Technology at the Jacob of Paradies University in Gorzow Wielkopolski (Gorzow Wielkopolski, Poland).

## 3. Results and Discussion

The selection of appropriate parameters has an influence on the attainment of a high-quality cutting surface. The basic machining parameters that have a decisive impact on shaping the geometry of the cutting gap include the value of water pressure, the speed of the cutting head feed, the type and amount of abrasive, the diameter and length of the nozzles, the distance of the cutting nozzle from the material being cut, and the properties and shape of the material being processed [35,36].

For the VIKOR method, calculations were made on the basis of the L9 decision matrix. Taking into account the important parameters of water cutting treatment as non-constant control values, the mass flow rate of the garnet was analyzed. The cutting depth and the roughness of the side surface of the cut groove were assumed as the starting point. The rest of the values of the control parameters involved in the process were considered unchanged. That were:-Water nozzle diameter: 0.3 mm,-Diameter of the water-abrasive nozzle focusing tube: 0.76 mm,-Nozzle-to-material distance: 4 mm.

The listed parameters are presented in Table 3. Roughness measurements were conducted on a 953 μm × 953 μm elemental surface using a Gauss filter with a 0.2 correlation threshold.

Figure 4 shows a visualization of the obtained results for cutting deph and roughness. 

The depth of cut is an example of a favorable starting parameter, and the roughness factor Sq is included in the unfavorable factor. The VIKOR approach has proven to be a beneficial method of optimizing the challenges of the water jet cutting process. This multi-characteristic approach significantly reduces the calculations and determines the positions from the calculation results by the optimal combination of control parameters. The VIKOR method used in the research effectively deals with contradictory answers. There are two criteria and nine alternatives in this study. Table 4 shows the calculated indexed entropy values, dispersion values, and weight vector [37].

Table 5 shows the decision matrix.

Table 6 shows the normalized decision matrix

Table 7 shows the values $S_{i}$, $R_{i}$ and *Q_i_*.

Figure 5 shows the distribution of S, R and Q values for individual alternatives. 

The VIKOR method made it possible to determine the values S, R, and Q for individual alternatives, on the basis of which the ranking of options was built. For the above results, numbers were assigned in the ranking, assuming that the highest place is occupied by the value closest to 0, and the lowest place by the value closest to 1. Table 8 contains a ranking list based on the obtained R, S, and Q values.

In order for a solution with an alternative A(1) to be considered a compromise by the value of Q (minimum), two criteria must be met [38]:-Condition 1. Advantage is accepted: Q(A(2) − Q(A(1) ≥ 1/(m^−1^) where A(1) is the alternative highest ranked, and A(2) is the next alternative after Q and m is the number of alternatives.-Condition 2. Stability is accepted. This means that the alternative A(1) must also meet the requirement of the highest grade of S and/or R.

If neither condition 1 nor condition 2 are achievable, the VIKOR method assumes one of the compromise solutions:-Solution 1. When Condition 1 is not met then the value of A(m) is determined by Q(A(m)) − Q(A(1)) < 1/(m^−1^) for the largest value of m (these alternatives are close to each other).-Solution 2. When Condition 2 is not met, the compromise is A(1) and A(2).-Solution 3. When Condition 1 and Condition 2 are not satisfied, the compromise solution is the smallest value of Q.

The results of the conditions survey are shown in Table 9.

Therefore, alternative 3 is selected as the final alternative.

The results presented in Table 10 allow for the conclusion that the optimal parameters of the abrasive jet-cutting process are:-Pressure is the most important factor and the smallest dispersion can be observed at 400 MPa;-The abrasive flow rate is a parameter that slightly affects the surface roughness;-The optimum traverse speed for minimum surface roughness.


The optimal parameters for a combination of parameters are:-Pressure = 400 MPa;-Abrasive flow rate = 450 g/min;-Traverse speed = 150 mm/min.

Figure 6 shows a view of the cutting surface, processed with optimal control parameters, from an SEM microscope. These are the parameters indicated as control for the best AWJ machining results. Therefore, the use of the VIKOR method made it possible to indicate the optimal input parameters in order to obtain the most favorable output parameters. The values of the control parameters were determined in this way; a cutting test was carried out and the surface roughness was measured in analogous areas. 

The results obtained from the tests are generally consistent with the results of other studies. In particular in terms of the impact of feed speed, as quoted by Perec et al. [39] and Spadlo [40], or the abrasive flow described by Mazurkiewicz [41], Balamurugan [42], and Valíček et al. [43]. This confirms previous observations [15,44] that the use of multi-criteria methods are important for calculating the key control parameters of the AWJ cutting process.

In the VIKOR method, a matrix with 9 variants was used in the calculations. The method determined the optimal combination of machining parameters given in Table 10. It was found that the changes in the machining effects are most influenced by the change in feed rate then the abrasive flow rate. The smallest effect is caused by a change in pressure. It has been found to be an effective process optimization method and can be used in surface engineering, turning and water jet machining, which confirms the findings of Kumar [45]. The next steps of this method follow the scheme [46]:-Determination of points: ideal and anti-ideal;-Calculation of the weighted average distance from the ideal point S_i_ and the maximum weighted distance from the ideal point R_i_ for each object;-Determination of the comprehensive Q_i_ index for each variant followed by the construction of three rankings based on the calculated values according to the principle: the lower the value of the index, the higher the position in the ranking;-Selecting the first variant from the Q_i_ ranking and comparing it with the variant immediately after it in this ranking; two conditions are checked at this stage—acceptable advantage and acceptable stability of the decision—on the basis of the information obtained. It is decided which variant or variants are compromise solutions.

The VIKOR method, like other multi-criteria methods, can be used to optimize new processes as well as to improve existing ones. VIKOR, like Taguchi or TOPSIS, is based on the concept of measuring the distance of the tested variant from the ideal scenario. This measurement is made using three metrics and additional conditions are considered later. Importantly, the VIKOR method has various variants, e.g., based on fuzzy set theory.

The main purpose of this method is to identify a compromise solution that ensures the maximum utility of favorable parameters (represented by min S) and the minimum utility of unfavorable parameters (represented by min R). The VIKOR method works well in experimental measurements of the degree of changes introduced to the process in its various phases [46] and reduces costs while improving the quality of the product or service in the modeling of the water jet cutting process in abrasive slurry [47], the water jet cutting process itself [48] and the effects of grinding wheel modifications [49,50], and even age hardening by shot blasting [51,52].

In order to select the parameters that will allow the achievement of optimal results, it is necessary to perform a number of actions and it is a laborious process. The use of the VIKOR method makes it possible to shorten the testing time, but it should be borne in mind that the correct application of this method requires the combination of knowledge from many fields and of areas of the company’s activity. Thanks to this, it is possible not only to optimize the process of cutting with a water-abrasive stream, but also to validate the entire production process, speed up the time of the process, and thus also save money for the company. Conducting the study using an IT system made it possible to obtain the desired result faster, i.e., such a ranking of input parameters not only had a significant impact on the titanium cutting process, but also made it possible to obtain a mathematical model and obtain relationships between input and output parameters. In addition, the obtained result gave the opportunity to specify the impact of specific input parameters on the expected result and to determine the impact of changing the input data size on the variability of the entire process.

## 4. Conclusions

The aim of this article was to present the results of research on the use of the VIKOR method in order to optimize the process of water-abrasive cutting of titanium alloy elements using the HPX garnet abrasive. A proprietary IT system was used as a supporting tool, which was tested in terms of efficiency in the optimization of the production process. By using the VIKOR method, it was possible to determine the optimal output parameters, i.e., the depth of cut and surface roughness. Furthermore, it was possible to determine the optimal input parameters, i.e., the values for abrasive flow rate, pressure, and traverse speed. Both input and output parameters are crucial and important for the water-abrasive cutting process. Based on the results obtained, it should be concluded that the VIKOR method can be used to determine the optimal parameters for abrasive jet cutting.

The VIKOR method is an effective tool that can be used to plan and carry out water-abrasive cutting processes. The output parameters obtained are consistent with the desired ones, i.e., the highest value for cutting depth and the lowest possible value for roughness. Most importantly, the optimized results, speed, and low complexity of the VIKOR method can be useful for testing new materials, including in the assessment of the impact of individual input and output parameters on their cutting with an abrasive water jet. The use of the VIKOR method in the proprietary IT system allowed for a faster solution to the problem of abrasive water jet cutting. When planning further research into the use of the VIKOR method, other input parameters should also be taken into account in the tests carried out, i.e., water nozzle diameter, nozzle-to-material distance or the use of other abrasive materials. The research of the effect of control parameters on the width of the cut kerf exceeds the framework of this work. The width of the cut groove does not directly affect the efficiency and quality of the cutting process, but the author intends to conduct such studies in further research.

The system examined selects the appropriate process parameters but also learns download data and results, and searches for historical and statistical data to support its decision. The system supports the entire production process from planning, technical data, investment availability, and production technology to delivery. The use of an IT tool as an auxiliary tool in the decision calculation process facilitated and streamlined the entire process.

## Figures and Tables

**Figure 1 materials-16-05405-f001:**
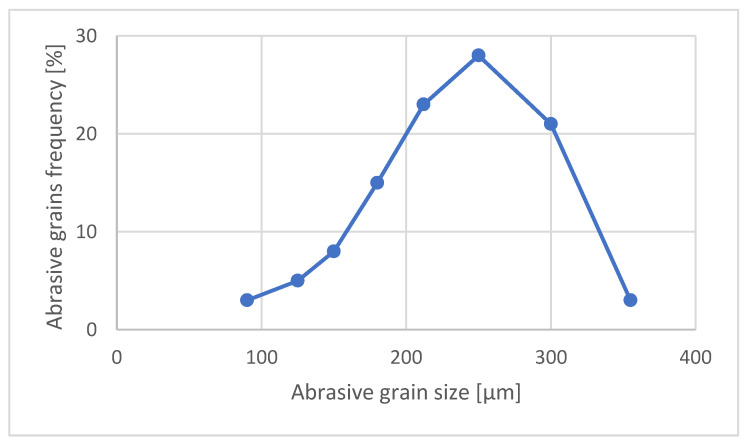
Typical grain size distribution of HPX80 garnet.

**Figure 2 materials-16-05405-f002:**
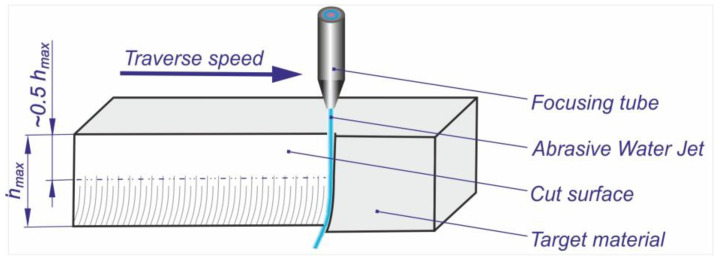
Schematic of cutting with AWJ.

**Figure 3 materials-16-05405-f003:**
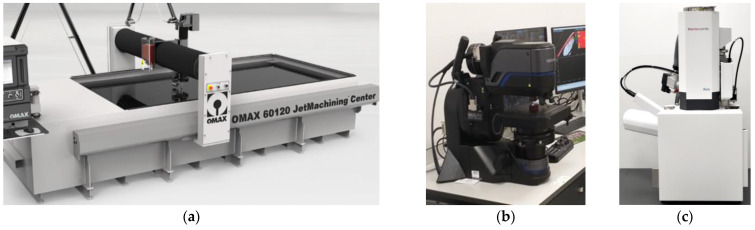
Research equipment: (**a**) OMAX 60120 JetMachining Center, (**b**) OLYMPUS DSX 1000 digital microscope, (**c**) Thermo Scientifi Axia ChemiSEM Scanning Electron Microscope.

**Figure 4 materials-16-05405-f004:**
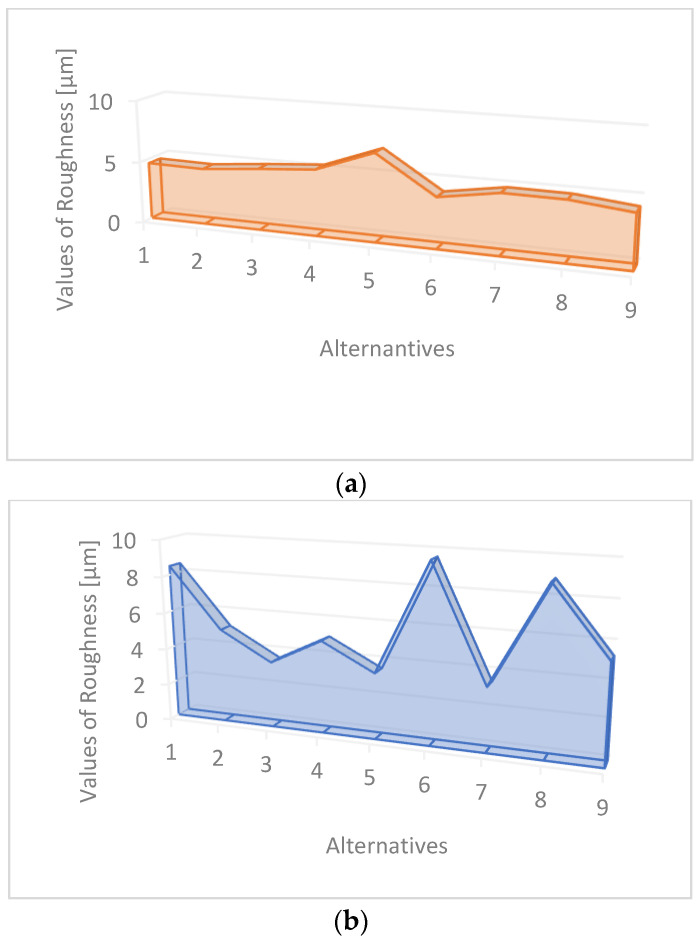
Visualization of the obtained results of cutting depth (**a**) and roughness (**b**) measurements.

**Figure 5 materials-16-05405-f005:**
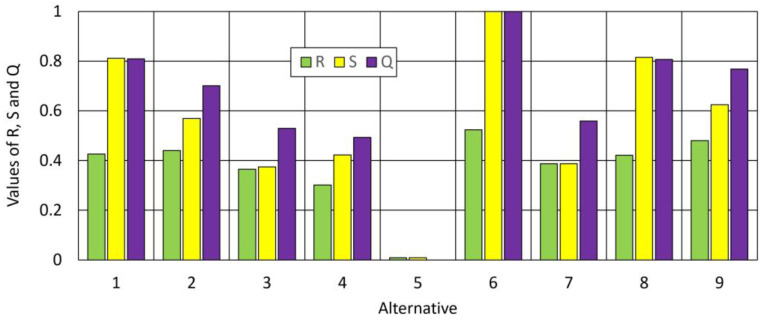
The distribution of S, R, and Q values for individual alternatives.

**Figure 6 materials-16-05405-f006:**
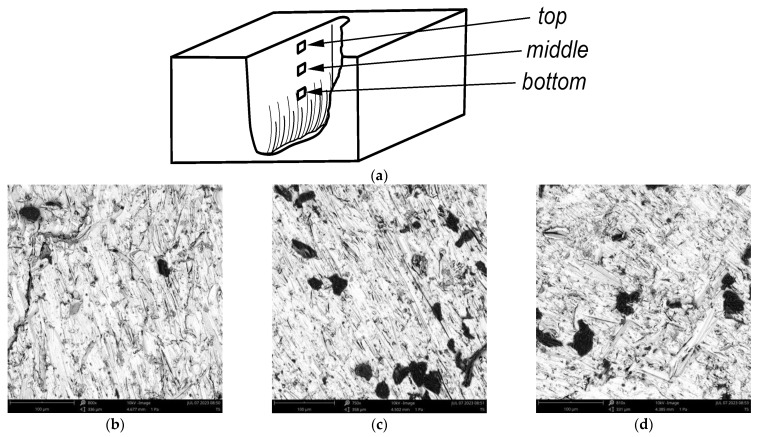
Sample SEM microscope view of cut surface details area with (**a**) areas locations, (**b**) top surface, (**c**) middle surface, and (**d**) bottom surface.

**Table 1 materials-16-05405-t001:** Typical composition of Grade 5 titanium [28].

Element	Contents [%]
Carbon	0.08 max
Nitrogen	0.05 max
Oxygen	0.20 max
Hydrogen	0.0125 max
Vanadium	3.50–4.50
Aluminum	5.50–6.75
Iron	0.25 max

**Table 2 materials-16-05405-t002:** Properties of Barton HPX garnet.

	Feature	Unit	Values
	Fe_3_Al_2_(SiO_4_)_3_+ Mg_3_Al_2_(SiO_4_)_3_+ Ca_3_Al_2_(SiO_4_)_3_	%	92.0–98.0
Chemical composition	Fe_3_O_4_+ NaCa_2_(Mg, Fe, Al)_3_(SiAl)_8_O_22_(OH)_2_+ KAlSi_3_O_8_+ (Ca, K, Na, Fe, Mg, Mn, Li, Al)_2-3_ (OH, F)_2_(Si, Al_4_O_10_)	%	4.0–8.0
	CaCO_3_	%	<0.2
	SiO_2_	%	<0.5
	ZrSiO_4_	%	<0.2
Melting temperature		°C	1315
Density		kg/m^3^	3900–4100

**Table 3 materials-16-05405-t003:** Input and output parameters of the VIKOR method.

No.	Input Parameters			Output Parameters	
				Beneficial	Non-Beneficial
	Abrasive Flow Rate [g/min]	Pressure [MPa]	Traverse Speed [mm/min]	Cutting Depth [mm]	Roughness Sq [mm]
1	250	360	50	8.47	4.64
2	250	380	150	5.16	4.56
3	250	400	250	3.61	4.99
4	350	360	150	5.05	5.35
5	350	380	250	3.59	7.07
6	350	400	50	9.64	4.08
7	450	360	250	3.48	4.86
8	450	380	50	8.93	4.82
9	450	400	150	5.35	4.33

**Table 4 materials-16-05405-t004:** Characteristics of criteria.

Criterium	Cutting Depth [mm]	Roughness Sq [μm]
entropy	0.537320329	0.608208812
dispersion	0.462679671	0.391791188
weight	0.54	0.46

**Table 5 materials-16-05405-t005:** Decision matrix.

	Beneficial	Non-Beneficial
	Cutting Depth	Roughness Sq
	[mm]	[μm]
alternative1	8.47	4.64
alternative2	5.16	4.56
alternative3	3.61	4.99
alternative4	5.05	5.35
alternative5	3.59	7.07
alternative6	9.64	4.08
alternative7	3.48	4.86
alternative8	8.93	4.82
alternative9	5.35	4.33

**Table 6 materials-16-05405-t006:** Normalized decision matrix.

	Beneficial	Non-Beneficial
	Cutting Depth [mm]	Roughness Sq [μm]
alternative1	0.444439	0.3072390
alternative2	0.270756	0.3019417
alternative3	0.189425	0.3304143
alternative4	0.264985	0.3542518
alternative5	0.188375	0.4681421
alternative6	0.505832	0.2701584
alternative7	0.182603	0.3218063
alternative8	0.468577	0.3191577
alternative9	0.280726	0.2867122

**Table 7 materials-16-05405-t007:** The values S, R, and Q.

	R	S	Q
alternative1	0.425849	0.811450	0.809727
alternative2	0.439869	0.569691	0.701410
alternative3	0.364513	0.374559	0.529915
alternative4	0.301424	0.422746	0.493022
alternative5	0.008500	0.008500	0
alternative6	0.523988	1	1
alternative7	0.387295	0.387295	0.558436
alternative8	0.421147	0.815452	0.807185
alternative9	0.480176	0.624680	0.768236

**Table 8 materials-16-05405-t008:** The ranking list for the alternatives.

	R Value	Rank in R	S Value	Rank in S	Q Value	Rank in Q
alternative1	0.425849	6	0.811450	7	0.809727	8
alternative2	0.439869	7	0.569691	5	0.701410	5
alternative3	0.364513	3	0.374559	2	0.529915	3
alternative4	0.301424	2	0.422746	4	0.493022	2
alternative5	0.008500	1	0.008500	1	0	1
alternative6	0.523988	9	1	9	1	9
alternative7	0.387295	4	0.387295	3	0.558436	4
alternative8	0.421147	5	0.815452	8	0.807185	7
alternative9	0.480176	8	0.624680	6	0.768236	6

**Table 9 materials-16-05405-t009:** Result of the conditions survey.

Condition 1	Acceptance
Condition 2	Acceptance
Selected solution	Solution 3

**Table 10 materials-16-05405-t010:** A optimal alternative.

No.	Input Parameters			Output Parameters	
				Beneficial	Non-Beneficial
	Abrasive Flow Rate [g/min]	Pressure [MPa]	Traverse Speed [mm/min]	Cutting Depth [mm]	Roughness Sq [mm]
5	350	380	250	3.59	7.07

## Data Availability

Not applicable.

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
