# Peer review of "Experimental Research on the Use of a Selected Multi-Criteria Method for the Cutting of Titanium Alloy with an Abrasive Water Jet"

_materials, 2023, doi:10.3390/ma16155405_

Round 1
Reviewer 1 Report
1In the Introduction, this paper only focuses on the advantages of VIKOR in multi-factor optimisation. Then in most papers, there are standard orthogonal methods, regression fitting, response surface methods and neural network algorithms with which this method can be compared in this paper. Reference can be made to the following papers:
2. what the test for weights is, and how to ensure the reliability of the values;
3 For Table 4, what is the basis of the experimental design?
4 Is the weight vector the degree of influence on the dependent variable
5 The study of each machining parameter's effect on the machined surface's quality needs to be more method-dependent and needs research on the cutting mechanism.
Author Response
Thank you very much for reviewing the article and any comments. Below are the answers to the comments.
- The literature review has been expanded with attention. Suggested papers are also included.
- Entropy Weight Determination approach was used to to determine the weights. In the entropy technique, the number of choices, and different criteria get to appraise multiple criteria optimization on basis establishing a comparative decision matrix as described Perec and Musial [A. Perec i W. Musial, „Multiple Criteria Optimization of Abrasive Water Jet Cutting Using Entropy-VIKOR Approach”, w Advances in Manufacturing Engineering and Materials II, S. Hloch, D. Klichová, F. Pude, G. M. Krolczyk, i S. Chattopadhyaya, Red., w Lecture Notes in Mechanical Engineering. Cham: Springer International Publishing, 2021, s. 50–62. doi: 10.1007/978-3-030-71956-2_5]
- The basis is an orthogonal table L9, selected for 3 parameters, assuming 3 levels each. In the the article also added a subsection Experiment Equipment, which describes the equipment on which the experiment was performed.
- The degree of weight does not affect on the dependent variable.
- To describe the process cutting mechanism in more detail, the article has added section Experimental setup and equipment.

Reviewer 2 Report
The research paper focuses on the application of the VIKOR method for selecting optimal parameters in abrasive cutting of titanium using a water jet. However, several key areas need improvement. Firstly, the paper lacks an investigation into the key areas of the cutting process and its impact on output responses, which would provide a deeper understanding of the underlying mechanisms. Additionally, the experimental setup is not adequately described, and there is a lack of discussion on the cutting process itself, which hinders the readers' comprehension.
Moreover, the paper lacks sufficient information about the testing conducted for Tables 1 and 2. Details regarding the specific experiments, including the methodology, equipment used, and measurement techniques, and some cutting analysis should be provided to ensure transparency and reproducibility of the study.
Furthermore, the introduction is brief and should be expanded to include a comprehensive review of the recent state-of-the-art in multi-objective optimization (MOO), a comparative analysis of different methods, and a clear rationale for selecting the VIKOR method, particularly in the context of machining titanium alloys.
Author Response
Thank you very much for your comments to the article. To increase its value, the article extended the introduction with a broader analysis of the problem. An Experimental research section has also been added, describing the equipment used to conduct the experiment.

Reviewer 3 Report
1. The “titanium” in the title is ambiguous, the pure titanium or titanium alloy material? It is better to further specify as “titanium alloy” or “Ti6Al4V titanium alloy”.
2. In table 1, the coal is the carbon element?
3. The experimental setup and procedure should be more detailed description, such as the nozzle, the cut process, it is better to provide some images of these information.
4. How about the effect of cutting parameters on the cutting width?
5. The author claimed that “As a result of the research, the best of the working nozzle dimensions and the level of abrasive flow rate were determined to obtain the greatest depth of cut”, but there is no change of nozzle dimensions in the paper, all nozzle dimensions are fixed in the experiments.
6. The theoretical analysis is too little in the results.
no
Author Response
Thank you very much for reviewing the article and any comments. Below are the answers to the comments.
- The description has been corrected to “Titanium alloy”
- The description has been corrected to “carbon”
- The article indicates: The rest of the values of the control parameters involved in the process were considered unchanged. These include:
- diameter of water nozzle: 0.3 mm,
- diameter of water-abrasive nozzle: 0.76 mm,
- distance of nozzle from material: 4 mm.
The article adds a subsection Experiment Equipment, which describes the equipment on which the experiment was performed.
- The research of the effect of control parameters on the width of the cut kerf exceeds the framework of the work. The width of the cut groove does not directly affect the efficiency and quality of the cutting process, but the author intends to conduct such studies in the near future, as signalled in the conclusions for further research.
- The description has been corrected in the article. Indicated: As a result of the conducted research, the best input parameters of the cutting process for abrasive flow rate, pressure and traverse speed were determined. The achieved result is consistent with the assumption that the most favorable output parameters are the highest cutting depth and the lowest roughness.
- In line with the remark, the literature review has been extended.

Round 2
Reviewer 2 Report
Improved and accepted in its current form.
Proof read and minor editing required.
Author Response
Good morning, thank you for your comments. I am sending a file with them in mind. Best regards

Reviewer 3 Report
All comments have been revised.
No.
Author Response
Good morning, I am attaching the corrected article, it also contains the improvement of the English language.
